# Adipose-Derived Stem Cell Exosomes as a Novel Anti-Inflammatory Agent and the Current Therapeutic Targets for Rheumatoid Arthritis

**DOI:** 10.3390/biomedicines10071725

**Published:** 2022-07-18

**Authors:** Ting-Hui Chang, Chien-Sheng Wu, Shih-Hwa Chiou, Chih-Hung Chang, Hsiu-Jung Liao

**Affiliations:** 1Department of Internal Medicine, Far Eastern Memorial Hospital, New Taipei City 220216, Taiwan; ahui815@yahoo.com.tw (T.-H.C.); wucs.tw@gmail.com (C.-S.W.); 2Department of Medical Research, Taipei Veterans General Hospital, Taipei 100136, Taiwan; shchiou@vghtpe.gov.tw; 3Genomic Research Center, Academia Sinica, Taipei 115021, Taiwan; 4Institute of Pharmacology, National Yang-Ming Chiao-Tung University, Taipei 300093, Taiwan; 5Department of Orthopedic Surgery, Far Eastern Memorial Hospital, New Taipei City 220216, Taiwan; 6Graduate School of Biotechnology and Bioengineering, Yuan Ze University, Taoyuan City 320315, Taiwan; 7Department of Medical Research, Far Eastern Memorial Hospital, New Taipei City 220216, Taiwan

**Keywords:** adipose-derived stem cell, rheumatoid arthritis, exosome, anti-inflammation

## Abstract

Patients with rheumatoid arthritis (RA), a chronic inflammatory joint disorder, may not respond adequately to current RA treatments. Mesenchymal stem cells (MSCs) elicit several immunomodulatory and anti-inflammatory effects and, thus, have therapeutic potential. Specifically, adipose-derived stem cell (ADSC)-based RA therapy may have considerable potency in modulating the immune response, and human adipose tissue is abundant and easy to obtain. Paracrine factors, such as exosomes (Exos), contribute to ADSCs’ immunomodulatory function. ADSC-Exo-based treatment can reproduce ADSCs’ immunomodulatory function and overcome the limitations of traditional cell therapy. ADSC-Exos combined with current drug therapies may provide improved therapeutic effects. Using ADSC-Exos, instead of ADSCs, to treat RA may be a promising cell-free treatment strategy. This review summarizes the current knowledge of medical therapies, ADSC-based therapy, and ADSC-Exos for RA and discusses the anti-inflammatory properties of ADSCs and ADSC-Exos. Finally, this review highlights the expanding role and potential immunomodulatory activity of ADSC-Exos in patients with RA.

## 1. Introduction

Rheumatoid arthritis (RA) is a chronic systemic autoimmune disease characterized by persistent synovial inflammation, which results in cartilage erosion and bone loss [1,2,3,4,5,6]. The mechanism underlying its development is highly complex because it involves an interplay between genetic and environmental factors [7]. The infiltration of inflammatory cells is a pathologic feature of RA, followed by synovial membrane hyperplasia and osteoclast overactivation [8]. Many studies have indicated that various types of immune cells, including T cells, macrophages, and B cells, are responsible for RA pathogenesis [9,10,11,12,13].

The main mechanism of action of the currently available RA therapeutics is inhibiting inflammation, to reduce joint damage and prevent bone loss. However, the conventional disease-modifying antirheumatic drugs (DMARDs), glucocorticoids, and nonsteroidal anti-inflammatory drugs (NSAIDs) used for RA treatment are nonspecific. Therefore, they non-selectively affect physiological pathways other than the immune response, increasing the risk of adverse events in the long term [14,15,16]. Several types of biologic DMARDs (bDMARDs) and small-molecule compounds exist, such as Janus kinase (JAK) inhibitors, compounds that target specific immune cells, cytokines, and compounds that modulate cytokine pathways [17,18]. In contrast to NSAIDs or glucocorticoids, which temporarily alleviate pain and inflammation, DMARDs delay RA progression [19]. DMARD treatment is effective in most patients with RA; however, 20%–30% of patients with RA do not respond to the treatment, or they experience major side effects from it. Moreover, biologics are not recommended for patients with compromised immune systems or infections because their use can increase the risk of infection [20].

Adipose-derived stem cells (ADSCs), which have strong immunomodulatory properties, can be readily isolated from adipose tissue and cultured in vitro [21,22,23]. Human ADSCs have been shown to be a promising therapeutic agent for RA [24]. Furthermore, they have a unique immunological profile, which enables adoptive transfer in allogeneic recipients with minimal risk of rejection. Therefore, ADSCs may have great therapeutic potential in patients with RA. As such, the establishment of cell banks using ADSCs from relatively few donors is warranted to facilitate widespread use among unrelated allogeneic recipients with RA [25,26,27]. It is important to note that patients with RA must receive medical treatments that may interfere with the outcomes of ADSC-based therapy. Therefore, the use of allogeneic ADSC-derived exosomes (Exos) may be easier than that of autologous mesenchymal stem cells (MSCs) or MSC-Exos in patients with RA.

Here, we comprehensively review the conventional and innovational treatment approaches used for RA treatment, such as MSC-Exo-based therapy, to understand the interplay between MSC-Exos and the innate and adaptive immune systems. Moreover, we discuss the biological characteristics of ADSC-Exos and review their experimentally verified RA therapeutic outcomes.

## 2. Current Therapy for RA

The currently available therapeutic drugs used for RA treatment can be classified into nonsteroidal anti-inflammatory drugs (NSAIDs), glucocorticoids, nonbiologic or conventional synthetic DMARDs (csDMARDs), biologic agents or bDMARDs, and targeted synthetic DMARDs (tsDMARDs) [28]. NSAIDs are most frequently used to mitigate pain. In addition, various combinations of corticosteroids may be administered to leverage their potent anti-inflammatory effects. Corticosteroids are potent anti-inflammatory drugs that modulate gene expressions by binding to glucocorticoid receptors to exert anti-inflammatory and immunosuppressive effects. However, their side effects include nausea, abdominal pain, osteoporosis, and metabolic disorders, which often worsen the quality of life of patients with RA [29].

csDMARDs are relatively safe and efficacious. Commonly used conventional DMARDs include methotrexate, sulfasalazine, leflunomide, and hydroxychloroquine [30]. In patients with definite RA, treatment with DMARDs is started as soon as the RA diagnosis is made. However, their use may be discontinued in the case of toxicity or loss of effect. In patients with well-controlled RA under csDMARD treatment, the discontinuation of the treatment increases the risk of flares. Patients started on DMARDs in the early stages of RA are likely to also require them in the later stages of the disease [31].

bDMARDs and tsDMARDs are added to RA treatment in patients unresponsive to tsDMARDs. bDMARDs (e.g., adalimumab, infliximab, certolizumab, and tocilizumab) are monoclonal antibodies that particularly target tumor necrosis factor (TNF)-α and interleukin (IL)-6 [32]. tsDMARDs typically target specific cytokine pathways; for instance, tofacitinib, baricitinib, filgotinib, upadacitinib, and decernotinib specifically target Janus kinase [33].

### 2.1. NSAIDs and Glucocorticoids

NSAIDs are ubiquitously used in rheumatology due to their effectiveness as anti-inflammatory and analgesic agents [34,35,36]. Although they differ widely in terms of chemical class, all NSAIDs can block prostaglandin (PG) production by inhibiting the enzyme PGG/H synthase, also called cyclooxygenase (COX). COX occurs in two isoforms, COX-1 and COX-2, which differ in regulations and tissue distribution and perform different biological functions: COX-1, expressed under basal conditions, is involved in PG production under homeostatic functions [37], whereas COX-2 expression increases during inflammation and other pathologic situations. The inhibition of COX-2 expression by NSAIDs blocks PG production at sites of inflammation or other forms of tissue damages [35]. However, due to low target specificity, COX-1 inhibition by NSAIDs in other specific tissues, most notably platelets and the gastroduodenal mucosa, can lead to common adverse effects such as bleeding and gastrointestinal ulceration [38]. Chronic systemic inflammation and cardiovascular risk factors are the main determinants of increased cardiovascular risk in patients with RA. However, NSAIDs may reduce the cardiovascular risk in patients with RA who frequently use them [39].

Glucocorticoids are anti-inflammatories with an essential role in RA management. Their clinical and structural efficacy has been widely acknowledged [40]. Most studies have evaluated their efficacy in bridging therapy, in combination with csDMARDs. However, most of the relevant studies have mainly focused on patients with early RA; there is a paucity of corresponding data for late-stage RA. Nevertheless, glucocorticoids may aid in controlling flares among patients with RA [41,42]. Adding glucocorticoids to bDMARDs and nonbiologic DMARDs most likely presents no or few undesirable effects, because most DMARDs have a fast onset of action [43,44]. Glucocorticoids are also inexpensive, and in combination with nonbiologic DMARDs, they may reduce or delay the need for the long-term use of biologic and nonbiologic DMARDs [45]. By contrast, glucocorticoid monotherapy is not acceptable because of the associated adverse effects [46]. Long-term, high-dose glucocorticoid use can lead to serious adverse effects, such as immunosuppression, osteoporosis, and metabolic disorders [47].

### 2.2. csDMARDs

csDMARDs, the cornerstone for RA management, can reduce pain and disability in patients with RA. However, all of the currently available csDMARDs have limited efficacy and some toxicity-related issues. Consequently, the interest in developing safe and effective treatment modalities for RA is growing. DMARDs are the immunosuppressive and immunomodulatory agents [48]. The most widely used csDMARDs include methotrexate, sulfasalazine, leflunomide, and hydroxychloroquine. Each DMARD has a unique mechanism of actions, but they all ultimately interfere with critical pathways in the inflammatory cascades. For example, methotrexate stimulates the releases of adenosine from fibroblasts, reduces neutrophil adhesion, inhibits leukotriene B4 synthesis by neutrophils, inhibits local IL-1 production, reduces IL-6 and IL-8 levels, suppresses cell-mediated immunity, and inhibits synovial collagenase gene expression [49]. Other medications in this class inhibit lymphocyte proliferation or cause lymphocyte dysfunction. For example, leflunomide suppresses dihydroorotate dehydrogenase, resulting in the inhibition of pyrimidine synthesis, thereby blocking lymphocyte proliferation [50]. Sulfasalazine has anti-inflammatory effects and prevents oxidative, nitrative, and nitrosative damages [51]. By contrast, hydroxychloroquine is an extremely mild immunomodulatory agent that blocks the intracellular toll-like receptor TLR9 [52].

### 2.3. bDMARDs

Numerous biological agents targeting various established cytokines or cellular subsets of the immune system have been developed. These agents demonstrate high efficacy for RA treatment; they can alleviate pain, improve quality of life, and considerably prevent structural damage. However, approximately 30%–50% of patients with RA do not demonstrate appropriate clinical responses to these drugs [53]. bDMARDs are highly specific and target a particular immune pathway. Some of these drugs are monoclonal, chimeric humanized fusion antibodies. Newer bDMARDs, with various modes of action, can be administered to RA patients who are unresponsive to TNF-inhibitor treatment [54]. For instance, abatacept (ABT, which interrupts the costimulation molecule signal between T and B cells) [55], rituximab (RTX, which targets CD20^+^ B cells) [56], and tocilizumab (TCZ, which is an IL-6 receptor (IL-6R) antagonist) [57] have demonstrated high treatment efficacy in patients with active RA, particularly in those with inadequate responses to TNF-inhibitor treatment [58].

### 2.4. Small-Molecule Compounds or tsDMARDs

Tofacitinib, the first FDA-approved tsDMARD, functions by inhibiting JAKs [59,60]. Other agents with similar modes of action are also available. However, these agents are generally recommended only to patients unresponsive to methotrexate or other conventional nonbiologic DMARDs or bDMARDs. All the JAK inhibitors used for RA treatment are administered orally; therefore, patients may prefer these agents over bDMARDs, which can only be administered intravenously or subcutaneously [61]. Small-molecule inhibitors have shorter half-lives than bDMARDs, and they need to be taken either once or twice a day. In general, small-molecule inhibitors are as small as ≤500 Da in size, much smaller than bDMARDs, which tend to be >1000 Da. tsDMARDs inhibit the intracellular signaling pathways of proinflammatory cytokines, in contrast to the mechanism of action of bDMARDs, which only block specific extracellular molecules [62,63]. Figure 1 summarizes the currently available bDMARDs and tsDMARDs for treating RA.

## 3. Generation of ADSCs and Their Therapeutic Application in RA

A large amount of adipose tissue can be obtained from the infrapatellar fat pads located in the extrasynovial area of the knee’s anterior compartment (Figure 2A) and subcutaneous fat (Figure 2B) removed through liposuction. ADSCs can exert immunosuppressive effects on both innate and adaptive immune cells, and they have been proposed as a therapeutic agent for autoimmune disease [64]. For instance, ADSCs have been used to modulate myeloid cells’ and lymphocytes’ differentiation toward their immunosuppressive phenotypes. The targeted immune cells include monocytes [65], macrophages [66], dendritic cells [67], and regulatory T (Treg) cells [68,69]. Furthermore, MSCs inhibit T and B cells’ proliferation and function [70,71,72].

MSCs derived from adipose tissue have therefore attracted considerable attention. Recent studies have evaluated the MSC dose tolerance, compared the efficiency of ADSCs and current RA drug treatments, investigated ADSCs’ capacity to alleviate or prevent the inflammation associated with RA and control other immune responses associated with RA propagation, and evaluated the safety of allogeneic ADSC transplantation [26,73]. The related clinical trials reported to date have consistently indicated that ADSCs are safe and effective in RA-related transplantation and treatment [74]. A phase Ib/IIa clinical trial showed that intravenously administered allogeneic ADSCs were safe and well-tolerated in patients with refractory RA, without dose-related toxicity for the dose ranges and exposure times studied [75]. In particular, three miRNAs—miR-26b-5p, miR-487b-3p, and miR-495-3p—were confirmed to be significantly upregulated in the ADSC-responder group compared to the ADSC-nonresponder group [25]. These miRNAs may represent novel candidate biomarkers associated with RA patients’ responses to ADSCs.

To enhance the understanding of the mechanisms underlying MSC-based RA treatment, the details of large-scale clinical trials with mandatory follow-up protocols reported to date are summarized in Table 1.

### 3.1. Clinical Studies of ADSCs in Patients with RA

Some patients with RA require long-term glucocorticoid treatment to control their RA symptoms. However, a Korean research team, for the first time, reported the outcomes of ADSC-based therapy and the subsequent discontinuation of glucocorticoid use in humans [76]. In the study, which included three patients with RA, ADSCs were extracted from autologous adipose tissues, cultured, expanded, and administered to patients intravenously or through intra-articular injection. The first patient, who received two intravenous injections of 3 × 10^8^ ADSCs, had improved pain visual analog scale (VAS) scores, which decreased from 10 to 2–3, and improved Korean Western Ontario and McMaster Universities arthritis index scores, which decreased from 73 to 28. The second patient received an intravenous injection of 2 × 10^8^ ADSCs and an intra-articular injection of 1 × 10^8^ ADSCs, followed by an additional intravenous injection of 3.5 × 10^8^ ADSCs and an intra-articular injection of 1.5 × 10^8^ ADSCs. This patient previously had difficulty walking, but after treatment, they could walk easily and stopped taking steroids. Finally, the third patient received four intravenous injections of 2 × 10^8^ ADSCs; after treatment, this patient could also walk easily and stopped taking steroids. This study indicates the potential efficacy and safety of autologous ADSCs in patients with RA. However, the limitations of the study were the small sample size, absence of objective indices with which to evaluate the response (e.g., the ACR response or DAS28 score), and short follow-up duration (3–13 months) [24,77,78].

The inflammatory environment of the RA synovium may trigger the immunomodulatory potential of ADSCs. Inflammatory synovial fluid from patients with RA can modulate ADSCs’ responses to induce Treg cells and modulate the phenotype of M2 macrophages. Proinflammatory synovial fluid maintains ADSC proliferation and upregulates the expression of the genes involved in immunomodulatory potential through a TNF- and NF-κB-dependent mechanism [79]. Furthermore, in a previous study, ADSCs exposed to proinflammatory synovial fluid were more effective in inducing Treg cells and inhibiting proinflammatory macrophages than those exposed to control synovial fluid, suggesting that maintaining ADSCs’ anti-inflammatory properties is essential for maximizing their effect in the local joint inflammatory environment.

The anti-inflammatory capacities of ADSCs from infrapatellar fat pads (IPFPs) and those from subcutaneous (SC) adipose tissues are the same. Skalska et al. reported that IPFP-MSCs and SC-MSCs obtained from patients with RA possessed similar immunomodulatory properties, despite their different localizations and the distinct cytokine milieus of the tissues of origin. Moreover, ADSCs derived from RA adipose tissues are not strongly immunosuppressive in vitro, and they may contribute to RA pathogenesis because of the enhancement of IL-17A secretion [74]. In their in vitro study, Skalska et al. reported that RA-ADSCs and osteoarthritis-ADSCs (OA-ADSCs) did not suppress the expression of activation markers in T cells or trigger Treg-cell expansion. Moreover, IPFP-MSCs from patients with RA and OA demonstrated comparable functions in vitro. However, their immunosuppressive activity appeared to be relatively impaired in patients with RA [26]. Table 2 demonstrates the potential of ADSCs and ADSC-Exos for alleviating the RA-related inflammation noted in vitro studies.

ADSC-based therapy has been reported to be efficacious, with great applicability in regenerative medicine. IPFP-MSCs, which can be easily extracted during orthopedic surgery, can resist inflammation and senescence, making them one of the best options for RA treatment. In addition, compared to bone-marrow-derived MSCs (BMSCs) and synovial MSCs, IPFP-MSCs possess a greater chondrogenic capacity with a unique age-dependent capacity for proliferation [82,83]. IPFP-MSCs have potential for ADSC-based RA treatment. Among all adult stem cells, SC-MSCs and BMSCs have identical properties [84] and are relatively accessible and abundant.

ADSCs also represent a potential alternative for MSCs in many therapeutic applications. The phenotype of an ADSC-Exo is similar to that of its parent MSC. In addition, ADSCs can provide more Exos than BMSCs or synovium stem cells. Adipose-cell-free Exos do not contain cells; thus, they cannot actively contribute to tumorigenesis, and they can be used for allogeneic transplantation. In addition, ADSC-Exos are easy to carry, transport, and store, with broad therapeutic potential [85]. ADSCs secrete several cytokines, growth factors, and antioxidant factors into their microenvironments and, thus, regulate the intracellular signaling pathways in the neighboring cells [86]. For the development of adipose-cell-free Exo-based RA therapy, the factors secreted from ADSCs and the ability to collect the abundant Exos are in high demand as a new treatment strategy. Several adipose-cell-free strategies for immunomodulation in RA are currently being explored.

ADSCs are abundant and can be easily collected through liposuction with minimal donor morbidity. In addition, compared to MSCs, allogeneic ADSCs exhibit improved anti-inflammatory capacity against RA because of the stronger genetic association of ADSCs with RA susceptibility and environmental risks. Until 2015, similar numbers of trials focused on allogeneic and autologous MSCs; however, more recent trials have focused on allogeneic MSCs rather than on autologous MSCs [87]. In particular, this trend indicates that future clinical trials in patients with autoimmune diseases may increasingly use allogeneic MSCs from cell banks established by private companies. Therefore, allogeneic MSC-Exos from non-RA donors may be useful for immunomodulation in synovial cells and, in turn, in the neighboring cells, thus promoting the resolution of arthritis. This approach represents a potential next-generation therapeutic strategy for RA.

### 3.2. Preclinical Studies on ADSCs in Animal Models

ADSCs have anti-inflammatory and regenerative properties. Several preclinical animal studies have investigated the effects of ADSCs in a mouse RA model and have identified the suppression of immune responses in vitro.

Ueyama et al. demonstrated that the localized injection of ADSCs and spheroids alleviated intra-articular inflammation and regenerated damaged cartilage in SKG mice with laminarin-induced arthritis. The authors also demonstrated the inhibitory effects of ADSCs and spheroids on synovial fibroblasts and activated macrophages through the elevation of *TSG6* and *TGFβ1* expression in vitro [23]. Moreover, metabolically engineered stem-cell-derived Exos systemically administered to mice with collagen-induced arthritis effectively accumulated in the inflamed joints and induced the anti-inflammatory activity through regulating macrophage phenotypes. In addition, the engineered Exos showed therapeutic efficacy equivalent to that of naked Exos at a 10-times-lower dose [88].

Gonzalez-Rey et al. reported that ADSCs stimulated the generation of FoxP3-expressing Treg cells, with the capacity to suppress collagen-specific T-cell responses. Finally, ADSCs downregulated the inflammatory responses and matrix-degrading enzyme production of synovial cells isolated from RA patients. The authors identified ADSCs as critical regulators of immune tolerance that can suppress T-cell and inflammatory responses and induce antigen-specific Treg-cell generation and activation [81]. Table 3 shows the potential of ADSCs and ADSC-Exos for alleviating the RA-related inflammation noted in preclinical animal studies.

## 4. Suppression of Joint Inflammation by MSC-Exos

### 4.1. Characteristics of MSC-Exos

MSCs, the cell type most studied in regenerative medicine, play a major role in tissue repair and generate local anti-inflammatory and tissue-healing signals [91,92]. MSCs elicit paracrine effects through extracellular vesicles (EVs). MSC-derived EVs, particularly Exos, have prominent therapeutic roles in disease-related tissue damage or inflammation.

Exos, which are 40–150 nm EVs formed by the fusion of multivesicular cell membranes, facilitate intercellular communication through the exchange of proteins, lipids, and RNA between cells [93]. Most Exos have a conserved set of proteins, including tetraspanins (CD81, CD63, and CD9), heat-shock proteins (HSP60, HSP70, and HSP90), the ALIX protein, and the protein encoded by tumor susceptibility gene 101 (TSG101); however, they also have unique tissue-type-specific proteins, which can reflect their cellular sources [94]. The characteristics of secreted Exos vary depending on their parent cells’ origin, type, and condition [95]. Phenotypically, MSC-derived EVs can express the MSC markers CD73, CD90, and CD105, but not CD14, CD34, or CD11b [96]. The functions of MSC-EVs are similar to those of their parent MSCs; however, the parent MSCs are more stable and safe and are less toxic, and they can more easily pass through the blood–brain barrier [97]. MSC-EVs transfer nucleic acids, lipids, proteins, and surface receptors from donor cells to specific recipient cells, thereby protecting signaling molecules from enzymatic degradation during transport [98]. MSC-EVs fuse with the recipient cell membranes by directly fusing with the plasma membranes, by fusing with the endosomal membrane after endocytosis, or by directly binding to the receptors of the recipient cells [99].

Similar to MSCs, MSC-Exos have anti-inflammatory and tissue-repairing effects, leading to a meaningful impact in RA treatment. MSC-Exos can also regulate RA symptoms through miRNAs and can reduce macrophage [88] and T- and B-cell [100] infiltration into RA cartilage. Therefore, MSC-Exos have various applications in regenerative medicine, including in immunomodulation [101] and cartilage repair [102]. They can be used as an alternative for cell-based therapy to maintain the characteristics of the original cells. The studies on the use of Exos in disease diagnosis and treatment to date have provided an essential basis for their future application in medicine.

### 4.2. ADSC-Exo Isolation and Identification

In RA therapy, MSC-Exos have more therapeutic potential than MSCs. First, allogeneic MSCs from patients with RA may have limited anti-inflammatory potential because of the intrinsic genetic and environmental risk factors. Therefore, whether autogenous MSC-EVs from patients with RA may secrete many other signaling molecules and trigger other diseases warrants further investigation. Second, MSCs may lead to tumor formation. Therefore, MSC-Exos are more effective, less toxic, and more stable than their parent cells [103]. Allogeneic MSC-Exos are regulated by specific pathways underlying RA symptoms. In addition, allogeneic MSC-Exos transfer various nucleic acids, proteins, and lipids from their parent cells to the recipient cells and thus participate in the inhibition of RA progression.

Notably, similar methods can be used for the isolation and characterization of Exos from different MSCs. In general, isolated Exos can be examined using transmission electron microscopy (TEM), nanoparticle tracking analysis (NTA), dynamic light scattering (DLS), resistive pulse sensing, and microfluidics and electrochemical biosensors [104,105]. MSC-Exos contain general markers, cell-surface markers (CD9, CD63, and CD81), transport proteins (annexins and flotillin), a heat-shock protein (HSP70), endosomal sorting complexes required for transport (ESCRT complex; TSG101, ALIX, and phospholipases), and other lipid-related proteins [106,107].

One of the most common methods used for Exo purification is differential ultracentrifugation (UC). In addition, an Exo precipitation–isolation method, which can be performed using a commercial Exo isolation kit, is commonly used. Tangential flow filtration (TFF) is another emerging technology that couples permeable membrane filtration and flow to obtain an efficient Exo concentration in a colloidal matrix. A comparative assessment of TFF and UC for a conditioned cell culture medium revealed that TFF aids in concentrating EVs with comparable physicochemical characteristics, with a high yield, few single macromolecules, and aggregates (size < 15 nm), and improved batch-to-batch consistency in half the processing time (1 h). Thus, Exos from adipose tissues are of high clinical relevance because they are expected to mimic the regenerative properties of their parent MSC cells.

Most relevant clinical studies have used different Exo preparation methods and have thus reported inconsistent results. Heterogeneous MSCs from patients with heterogenous RA demonstrate considerably diverse immunomodulatory and differentiation functions. Therefore, the types of molecules assembled by the extracted Exos may be heterogenous and may show altered functions in recipient cells, thus eliciting changes in the physiological processes. Moreover, miRNAs do not enter MSC-EVs equally. Finally, MSC-Exo separation methods have not been standardized to date. Even when commercial Exo extractants are used, the resulting Exos may contain non-Exo contaminants, such as lipoproteins, warranting the further purification of said Exos [103]. Therefore, the findings presented here should be replicated in large-scale clinical trials to assess the safety, effectiveness, and persistence of allogenous MSC-Exos in patients with RA.

### 4.3. MSCs and MSC-Exos in Animal Models

MSCs mainly interact with both innate and adaptive immune cells to modulate immune responses in patients with RA. MSC-based therapy is currently administered to patients with RA unresponsive to conventional RA therapy, and it is not associated with serious adverse events. Although several clinical trials have reported the effects of MSC-based therapy in patients with RA, no optimal MSC-based therapeutic protocol for patients with RA has been proposed [108]. Therefore, the use of MSCs in RA treatment is challenging. Most studies have used allogeneic MSCs, because isolating and propagating a sufficient number of autologous MSCs from patients with RA can be difficult. Autologous MSCs from patients with RA may have intrinsic genetic defects that may impair their anti-inflammatory capacity. Notably, patients with RA require long-term treatment to control symptoms and prevent structural damage, which may hamper MSCs’ function.

Accumulating evidence suggests that MSC-Exos efficiently transfer small-molecule drugs or proteins to target cells. However, MSC-Exos have not been applied to RA treatment. Nevertheless, the effectiveness of MSC-Exo-based therapy has been established in experimental animal models of RA, where MSC-Exo-based therapy has been noted to significantly alleviate the induction and progression of experimental arthritis. Of these models, the collagen-induced arthritis (CIA) mouse model is most widely used for assessing the efficacy of MSC-Exo-based therapy. Other studies have shown that MSC-Exos can suppress inflammation more effectively in RA than can MSC-derived microparticles [100]. MSC-Exos were noted to suppress T-cell proliferation in a dose-dependent manner and lower the numbers of mature T- and B-cell subsets [100,109,110]. Exos more effectively enhanced the population of Treg cells than did MSCs [110] and MSC-derived microvesicles [100]. However, some studies have demonstrated that microvesicles derived from MSCs are less effective in inducing TGFβ and IL-10 production in B and T cells than are MSCs alone [111]. Li et al. reported that, in CIA mice, MSC-derived miRNA-150-5p-expressing Exos reduce the secretion of inflammatory cytokines, including TNF-α and IL-1β, suppressing RA progression in vivo [112]. Moreover, miR-146a-transduced MSC-Exos were found to increase *FoxP3* (a transcriptional factor expressed in Treg cells), *TGFβ*, and *IL10* expression in CIA mice [90]. MSC-derived exosomal circFBXW7 suppressed the proliferation, migration, and inflammatory responses of rheumatoid fibroblast-like synoviocytes and alleviated RA in rats by sponging miR-216a-3p and activating HDAC4 [113].

Studies have also shown that targeted alterations of gene expression in Exos can alleviate RA by enhancing the Exos’ anti-inflammatory capacity. Tsujimaru et al. demonstrated that ADSC-Exos were themselves involved in RA-amelioration mechanisms by examining the functional effects of ADSC-Exos in RA development. Moreover, in mice, wild-type ADSCs and ADSC-Exos alleviated joint inflammation more significantly than did IL-1ra^–/–^ mouse ADSC-Exos [89]. These promising results have paved the way for MSC-Exo-based therapy to become a favorable new treatment modality for RA in humans.

Exo-transfected miRNAs could alleviate inflammation in mice with RA. Tavasolian et al. reported that, in mice with CIA, miR-146a-transduced ADSC-Exos increased *FoxP3*, *TGFβ*, and *IL10* expression, whereas miR-155-transduced ADSC-Exos increased *RORγt*, *IL17*, and *IL6* expression. Therefore, Exos can act as vehicles for the intracellular transfer of miRNAs between cells and represent a possible therapeutic strategy for RA [90]. Other in vitro studies showed increased target miRNAs in Exos derived from bone marrow stem cells, inhibiting inflammation in mice with RA; these miRNAs included miR-150-5p, miR-548e, miR-34a, miR-320a, miR-124a, miR-216a-3p, miR-192-5p, and miR-143-3p [111,113,114,115,116,117,118,119].

Figure 3 illustrates the currently available therapies for RA, including cell and non-cell therapies. To date, human studies on the effects of ADSCs on RA have been limited by small sample sizes, a lack of objective indices for response evaluation, and short follow-up durations [24,77,78]. Therefore, the interpretation of the anti-inflammatory potential of autogenous ADSCs is difficult. Many recent studies have demonstrated that non-cell Exos can be stable and effective vehicles for specific proteins, lipids, and genetic materials, including mRNAs, miRNAs, other small noncoding RNAs, and DNA; therefore, they may be promising tools for drug delivery to target tissues or organs [120]. MSC-Exo miRNAs regulate many physiological processes at the transcriptional and posttranscriptional levels. Chen et al. reported that miR-150-5p-overexpressing MSC-Exos reduced joint destruction by inhibiting the hyperplasia of synoviocytes and angiogenesis in patients with RA. miRNA-carrying MSC-Exos may be a new strategy for stem-cell-derived drug and miRNA delivery in patients with RA [111].

### 4.4. Biomaterial and Clinical Applications of ADSC-Exos

In addition, although clinical-grade Exos can be produced using biomaterials and standard operating procedures, methods for the large-scale production of Exos for clinical use have not been reported. Nevertheless, valuable biomaterials can be used to maximize the therapeutic power of ADSC-Exos. Consequently, hydrogels have attracted considerable attention as biocompatible auxiliary materials in recent years [121].

In the RA experimental models, using an injectable hydrogel prolonged the retention time for the MSC-Exos, enhancing their therapeutic effect. The route and dose of Exo administration must thus be explored to ensure the safety of patients treated with ADSC-Exos. The primary administration route used in preclinical animal studies has been intra-articular. Therefore, the most appropriate route of Exo administration in RA patients in the clinic warrants investigation.

The methods for coating Exos on nanoparticle surfaces are similar to those for coating them on cell membranes. To date, most nanoparticles coated with Exos have been metallic, including gold nanoparticles, iron oxide nanoparticles (IONs), and gold-IONs. However, in some cases, poly(lactic-co-glycolic acid) nanoparticles and metal–organic framework nanoparticles have been coated with Exos. Exo-based nanoparticle coatings enable the expansion of nanoparticle targeting from typical approaches that target individual antigens to a strategy that can simultaneously target many antigens for more efficient uptake by target cells. In addition, Exo-derived coatings have immune-evasive properties that can increase the retention times [122]. Clinical trials monitoring patients treated with ADSC-Exos in real time are required to determine the effectiveness of ADSC-Exos at low doses. Therefore, future breakthrough research should combine basic research on ADSC-Exos with other emerging biomaterials for RA treatment.

## 5. Conclusions

RA treatment is challenging. In some patients with RA, the currently available treatment modalities cannot completely alleviate RA symptoms. In this review, we discussed the immunomodulatory effects of ADSCs and ADSC-Exos on immune cells, as well as RA progression under ADSC-Exo-based treatment; in general, ADSC-Exos may represent novel therapeutic agents for the cell-free or cell-component-based treatment of RA. Despite the encouraging results, denoted by target gene or miRNA overexpression, the clinical potential of ADSC-Exos as drug vectors in an appropriate form for administration warrants further analysis.

## Figures and Tables

**Figure 1 biomedicines-10-01725-f001:**
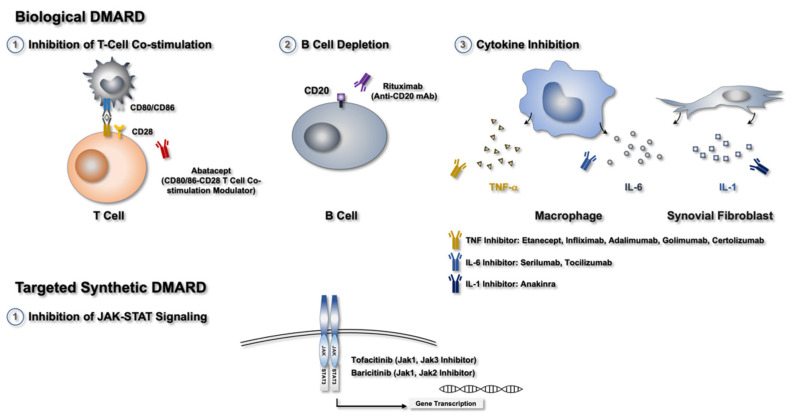
Mechanism of the currently available drugs for RA treatment.

**Figure 2 biomedicines-10-01725-f002:**
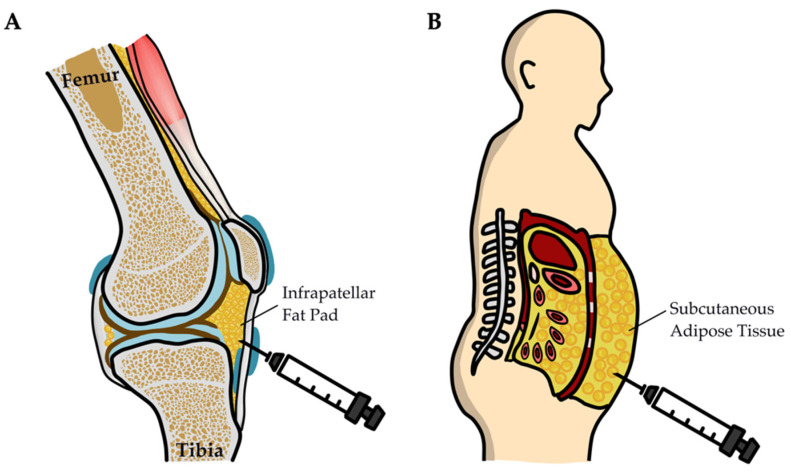
Infrapatellar fat pads (IPFPs) and subcutaneous fat pads, which are primary sources of ADSCs for RA treatment. (**A**) The IPFP is deep into the patella and locates the space between the patellar tendon, femoral condyle, and tibial plateau and plays a crucial biomechanical role within the knee. IPFP also acts as a reservoir of ADSCs. (**B**) The subcu-taneous fat pad represents an emerging alternative source of ADSCs and can be obtained from abundant adipose tissue by a minimally invasive procedure.

**Figure 3 biomedicines-10-01725-f003:**
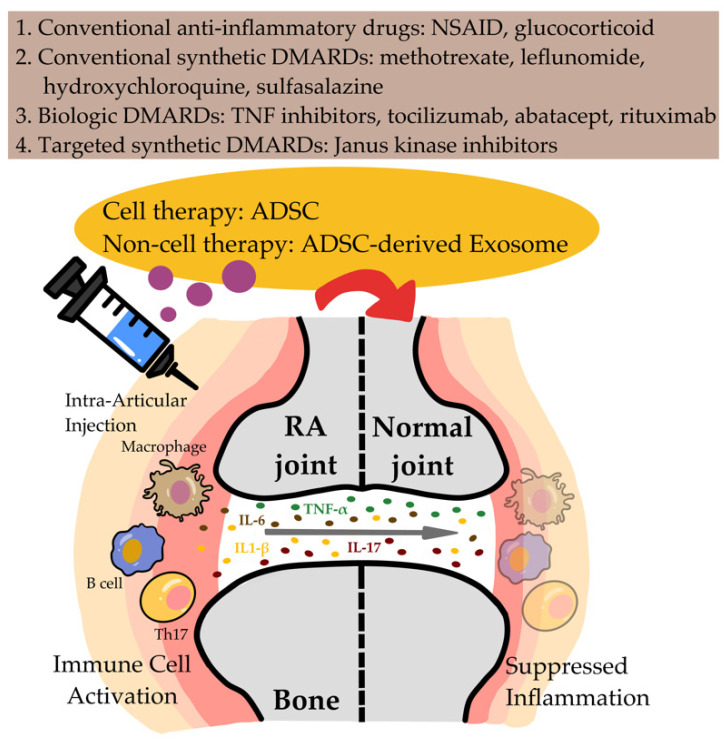
Schematic of the current therapies (drug, cell, and non-cell) used for RA treatment. ADSC-Exos mediate cell–cell communication and have immunosuppressive and immunomodulatory effects on adaptive and innate immunity, inhibiting bone destruction and promoting joint regeneration.

**Table 1 biomedicines-10-01725-t001:** Potential of ADSCs and ADSC-Exos for the alleviation of the RA-related inflammation noted in clinical studies.

ADSC Administration in Patients With RA				
Cell Source	Treatment Conditions	Outcome	Number of Patients	Reference
Clinical Study				
Allogeneic subcutaneous adipose tissue	(3–12) × 10^6^ cells/kg, i.v.	Treatment was generally well-tolerated, without dose-related toxicity in the dose range and time	46 (RA)	[75]
Autologous subcutaneous adipose tissue	(1.5–3.5) × 10^8^ ADSCs/kg, s.c.	Pain VAS and KWOMAC decreased, and walking improved	3 (RA)	[76]
Allogeneic subcutaneous adipose tissue	Expanded allogeneic ADSCs in refractory RA, i.v.	Three miRNAs, namely, miR-26b-5p, miR-487b-3p, and miR-495-3p, were significantly upregulated in the responder group (reduced MRI score) compared to the nonresponder group	14 (RA)	[25]

**Table 2 biomedicines-10-01725-t002:** Potential of ADSCs and ADSC-Exos for the alleviation of the RA-related inflammation noted in in vitro studies.

In Vitro Study				
Cell Source	Treatment Conditions	Outcome	Number of Patients	Reference
ADSC				
Subcutaneous adipose tissue	ADSCs first treated with SF and ADSC proliferation followed by gene expression of immunomodulatory factors	Conditioning ADSCs with proinflammatory RASF enhanced their ability to induce Treg cells and inhibited the proinflammatory markers CD40 and CD80 in activated macrophages	8 (RA)	[79]
Subcutaneous adipose tissue	ADSC–PBMCs cocultured with PMA treatment	ADSCs greatly upregulated Th2- and Treg-cell transcription factors (i.e., GATA3 and Foxp3) and downregulated Th1 and Th17 transcription factors (i.e., T-bet and RORγt)	14 (RA)	[80]
Infrapatellar fat pad or subcutaneous adipose tissues	PBMCs stimulated with PHA cultured alone or in the presence of naïve or TNF/IFNγ-pretreated ASCs isolated from infrapatellar fat pads or subcutaneous adipose tissues	IPFP-MSCs and SC-MSCs obtained from patients with RA had similar immunomodulatory properties despite the different localization and distinct cytokine milieus of the tissues of origin	8 (RA)	[74]
Infrapatellar fat pad	PBMCs from healthy donors cocultured with ADSCs from patients	The immunosuppressive properties of RA-ADSCs and OA-ADSCs were impaired	29 (RA) 12 (OA)	[26]
Subcutaneous adipose tissue	ADSCs from healthy donor cultured with collagen-reactive RA human T cells	ADSCs stimulated the generation of FoxP3 protein-expressing Treg cells, with the capacity to suppress collagen-specific T-cell responses from patients with RA	22 (RA, PBMC)	[81]

**Table 3 biomedicines-10-01725-t003:** Potential of ADSCs and ADSC-Exos for the alleviation of the RA-related inflammation noted in preclinical animal studies.

Preclinical Animal Study				
Cell Source	Treatment Conditions	Outcome	Animal Model/Species	Reference
ADSC				
Autologous subcutaneous adipose tissue	1.5 × 10^4^ ADSCs/knee, intra-articularly	Localized injection of ADSCs and spheroids reduced intra-articular inflammation and regenerated damaged cartilage in a mouse model of RA	Laminarin-induced arthritis/ SKG mice	[23]
ADSC-Exos				
Human ADSC cell line	10^7^–10^8^ dibenzocyclooctyne (DBCO)-conjugated dextran sulfate (DS)-conjugated ADSC-Exos, i.v.	DS-Exos systemically administered to mice with collagen-induced arthritis effectively accumulated in the inflamed joints, inducing a cascade of anti-inflammatory activity via regulation of macrophage phenotypes	Collagen-induced arthritis/ DBA-1J mice	[88]
Subcutaneous adipose tissue	5 mg EVs or 1 × 10^6^ ADSCs, i.v.	ADSC-Exos alleviated RA via transfer of factors such as IL-1ra	Collagen-induced arthritis/ BALB/c mice	[89]
Subcutaneous adipose tissue	Exos extracted from normal MSCs with overexpressed miR-146a and miR-155	Treatment with MSC-Exos and miR-146a/miR-155-transduced MSC-Exos significantly altered CIA mice’s Treg-cell levels and suppressed inflammation	Collagen-induced arthritis/ DBA-1J mice	[90]

ADSC, adipose-derived stem cell; Exo, exosome; RASF, RA synovial fluid; Treg, regulatory T cell; PBMC, peripheral blood mononuclear cell; PHA, phytohemagglutinin; IPFP-MSC, infrapatellar fat pad MSC; SC-MSC, subcutaneous MSC; s.c., subcutaneously; i.v., intravenously.

## Data Availability

Not applicable.

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
