# Peer review of "Adipose-Derived Stem Cell Exosomes as a Novel Anti-Inflammatory Agent and the Current Therapeutic Targets for Rheumatoid Arthritis"

_biomedicines, 2022, doi:10.3390/biomedicines10071725_

Round 1
Reviewer 1 Report
The authors of this review are summarising the use of adipose derived mesenchymal stem cells exosomes for the treatment of rheumatoid arthritis. They make comparisons with current therapies for rheumatoid arthritis, including NSAIDs and various DMARDs. They provide a summation of the treatment of rheumatoid arthritis using ASC exosomes applied both in animal models and a few clinical studies.
There remain open questions that the authors need to address prior to further review consideration.
1) The summary of current therapies is a nice overview. However, can the authors present a figure summarising the actions of these drugs (potentially two subfigures) to show their mechanism of action. This would allow a comparison with ASC exosomes that are discussed later in the review.
2) In the review of the literature on ASCs and their subsequent exosomes, are there differences in their secretomic action related to their origin (i.e. between knee or abdomen) or processing that could affect their potential action ? A comment needs to be made in this review.
3) How were the exosomes characterised in these studies ? Was there consistency in their characterisation and thus could be differences between studies be related processing or origin ? It is known that many clinical studies have inconsistencies between preparations that lead to differing results. A critical commentary needs to be added to the review.
4) Figure 2 needs to be significantly improved to provide a better summary of the mechanism of action between ASC exosomes therapies on joint inflammation. This could be compared to figure requested in qu. 1 to help readers get a greater understanding of rheumatoid arthritis treatment via ASC exosomes.
5) Table 1 should be separated between clinical and preclinical studies. Additionally, what animal were used in each of the preclinical studies ? This should be added to the table for a better comparison between studies.
6) What was the rationale to investigating ASC exosomes compared to bone marrow or synovium derived exosomes ? Apart from ease of extraction, are there differences in their efficacy and secretomic factors for the treatment of rheumatoid arthritis ?
Reviewer 2 Report
This is an interesting paper showing a possiblity that Adipose-Derived Stem Cell Exosome can act as a Novel Anti-Inflammatory Agent for Rheumatoid Arthritis.
According to the title, main issue of this review paper looks on exosome. However, only one section allowed the description on exosome. Other parts have been already reported by previous review papers.
Therefore, this paper looks lack of contents and still needs more contents on exosomes.
Round 2
Reviewer 1 Report
The authors have answered my questions in an appropriate manner.
Reviewer 2 Report
Title should be modified to include whole content of this paper.
Since this review explain not only other treatment cases but also Adipose-Derived Stem Cell Exosome as a Novel Anti-Inflammatory Agent for Rheumatoid Arthritis.
